# Role of Polyphenol-Derived Phenolic Acid in Mitigation of Inflammasome-Mediated Anxiety and Depression

**DOI:** 10.3390/biomedicines10061264

**Published:** 2022-05-28

**Authors:** Ruth Iban-Arias, Maria Sebastian-Valverde, Henry Wu, Weiting Lyu, Qingli Wu, Jim Simon, Giulio Maria Pasinetti

**Affiliations:** 1Department of Neurology, Icahn School of Medicine at Mount Sinai, New York, NY 10029, USA; ruth.iban-arias@mssm.edu (R.I.-A.); m.sebastian.valverde@gmail.com (M.S.-V.); henryvvu@gmail.com (H.W.); 2New Use Agriculture and Natural Plant Products Program, Department of Plant Biology, Center for Agricultural Food Ecosystems, Institute of Food, Nutrition & Health, SEBS, Rutgers University, 59 Dudley Road, New Brunswick, NJ 08901, USA; weitinglyu96@gmail.com (W.L.); qlwu@sebs.rutgers.edu (Q.W.); jimsimon@rutgers.edu (J.S.); 3Department of Medicinal Chemistry, Ernest Mario School of Pharmacy, Rutgers University, 160 Frelinghuysen Road, Piscataway, NJ 08854, USA

**Keywords:** malvidin glucoside, anthocyanin, metabolite, therapeutic, innate immunity, neuroinflammation, stress-related illness

## Abstract

Overexposure to mental stress throughout life is a significant risk factor for the development of neuropsychiatric disorders, including depression and anxiety. The immune system can initiate a physiological response, releasing stress hormones and pro-inflammatory cytokines, in response to stressors. These effects can overcome allostatic physiological mechanisms and generate a pro-inflammatory environment with deleterious effects if occurring chronically. Previous studies in our lab have identified key anti-inflammatory properties of a bioavailable polyphenolic preparation BDPP and its ability to mitigate stress responses via the attenuation of NLRP3 inflammasome-dependent responses. Inflammasome activation is part of the first line of defense against stimuli of different natures, provides a rapid response, and, therefore, is of capital importance within the innate immunity response. malvidin-3-*O*-glucoside (MG), a natural anthocyanin present in high proportions in grapes, has been reported to exhibit anti-inflammatory effects, but its mechanisms remain poorly understood. This study aims to elucidate the therapeutic potential of MG on inflammasome-induced inflammation in vitro and in a mouse model of chronic unpredictable stress (CUS). Here, it is shown that MG is an anti-pyroptotic phenolic metabolite that targets NLRP3, NLRC4, and AIM2 inflammasomes, subsequently reducing caspase-1 and IL-1β protein levels in murine primary cortical microglia and the brain, as its beneficial effect to counteract anxiety and depression is also demonstrated. The present study supports the role of MG to mitigate bacterial-mediated inflammation (lipopolysaccharide or LPS) in vitro and CUS-induced behavior impairment in vivo to address stress-induced inflammasome-mediated innate response.

## 1. Introduction

The high prevalence of stress is known to affect people globally and is considered detrimental to health and well-being when it becomes chronic or mismanaged [1]. When the stress stimuli are under control, an appropriate physiological response is elicited as a mechanism of defense. This response is known as allostasis, and it is beneficial and necessary for survival and recovery [2]. However, when the body feels continuously challenged and/or threatened by overexposure to a wide array of stressors, a pathophysiological response is induced that may lead to or worsen medical conditions of different natures [3]. Psychological stress represents a risk factor in 75% to 90% of diseases, including psychiatric disorders [4], cardiovascular diseases [5], metabolic diseases [6,7], and neurodegenerative diseases [8]. Evidence indicates that exposure to chronic stress can induce central and peripheral inflammation [9,10]. The systemic or chronic low-grade inflammation may lead to stress-related illnesses such as depression [11,12]. It has been reported that patients suffering from depression present elevated serum levels of pro-inflammatory cytokines including interleukins, e.g., IL-1β, IL-6, and tumor necrosis factor-alpha (TNF-α) [13,14]. The persistent activation of an inflammatory immune response may promote neurotransmitter imbalance in the brain and induce somatic changes that can translate into behavior impairment [4,15,16]. Microglia, the resident immune cells of the central nervous system, become activated in the brain upon psychological stress exposure, leading to a morphological change characterized by the acquisition of an amoeboid phenotype [17]. As a response to microglia activation, there is a recruitment of peripheral monocytes and macrophages that potentiates the circulating levels of pro-inflammatory cytokines [18]. Constant exposure to stressors over-activates the immune system, leading to homeostasis disruption by the release of pro-inflammatory cytokines including C-reactive protein, IL-6, TNF-α, IL-1β, and the transcription factor of nuclear factor kappa B (NF-κβ) [19]. Inflammasomes are cytoplasmic multimeric protein complexes essential for the first response of the innate immune system following exogenous or endogenous stimuli; therefore, they represent the first mechanism of defense against stressors of different natures [20,21]. Inflammasomes contain essential components such as the pattern-recognition receptors (PRRs), including the nucleotide-binding domain leucine-rich repeat-containing proteins (NLRs) and the absent in melanoma 2-like receptors (AIM2), which recognize pathogen-associated molecular patterns (PAMPs), derived from foreign pathogens, and danger-associated molecular patterns (DAMPs), induced by endogenous stress [22]. Included in the NLR, the nucleotide binding-domain leucine-rich repeat-containing protein 3 (NLRP3) and the NLR family CARD domain containing 4 (NLRC4) are the most well-characterized [23,24]. Activation of the NLRP3 inflammasome takes place in two stages: first, it must be transcriptionally primed, increasing its expression, to respond to stimuli, activate, and assemble [25]. The binding of LPS to the toll-like receptor 4 (TRL4, which belongs to the PRR family) can be a signal for NLRP3 for priming and ATP for activation. Subsequently, NLRP3 inflammasome oligomerizes and recruits pro-caspase-1 via adaptor molecule apoptosis-associated speck-like protein containing a CARD (ASC), leading to the cleavage of the pro-form of the IL-1 family, such as IL-1β, into their bioactive forms [26,27]. NLRC4 is known to respond to a more restricted set of stimuli than NLRP3 [28,29]. In general, inflammasome assembly propagates caspase-induced inflammation by IL-1β activation [30], leading to a type of highly inflammatory programmed cell death also known as canonical pyroptosis [31]. Pyroptosis is instrumental in the clearance of foreign pathogens and plays a major role in innate immunity and inflammatory-like diseases [32,33]. This process requires cleavage and activation of Gasdermin-D, a pro-pyroptotic protein effector, by human and mouse caspase-1, human caspase-4 and caspase-5, and mouse caspase-11 to form pores on the plasma membrane allowing the water to enter, resulting in cell swelling, osmotic lysis, and, eventually, rupture of the cell membrane and leakage of cytokines such as IL-1β [34].

Stress-induced inflammation is considered a disease risk factor; therefore, it represents an important target to treat inflammatory-related diseases. Since inflammasome assembly is crucial in the progression of the stress-induced inflammatory response, therapies to halt this process would be highly beneficial [35]. Large bodies of evidence indicate the potential therapeutic effect of polyphenols, present at different proportions in a variety of colored fruits, to contribute to the prevention and/or treatment of many diseases associated with chronic or systematic inflammation. Grape seed polyphenolic extract has been reported to exhibit a wide range of bioactivities including antioxidant [36], anti-diabetic [37], anti-proliferative, and breast cancer protection [38], among others. Furthermore, in vitro [39,40,41] and in vivo [42] studies support the ability of grape seed polyphenols to intervene in the Aβ and tau neuropathology, suggesting its potential role in the prevention and treatment of Alzheimer’s disease. Anti-inflammatory activity has also been validated in these natural compounds. Huang et al. [43,44] demonstrated the beneficial effect of the two most abundant anthocyanins in blueberry fruits, namely malvidin-3-glucoside (MG) and malvidin-3-galactoside on the endothelial cell-mediated inflammatory response. Another study showed that phenolic fractions from blueberry extract suppress inflammatory markers in macrophages in vitro, MG being one of the most effective [45]. MG is one of the most widespread anthocyanins and has been reported with an array of health-promoting beneficial actions, including anti-inflammatory. The present study supports the role of MG to modulate NLRP3, NLRC4, and AIM2 inflammasome activation, both in primary cortical microglia and in a mouse model of anxiety and depression, as a potential therapeutic approach to addressing immune-inflammatory-based disorders.

## 2. Materials and Methods

### 2.1. In Vitro Experiments

#### 2.1.1. Preparation of Murine Primary Cortical Microglia Culture

Mixed murine cortical cultures were prepared following previously described protocols with slight modifications [46]. In brief, cortices from wild type C57BL/6J p2-p3 pups were isolated, digested, and seeded in poly-D-lysine coated (5 µg/mL) 24-well plates at a density of 16 cortices per plate. Every three days, the medium (DMEM, 10% FBS, 1% penicillin-streptomycin) was replaced. After 21 days, mixed glial cultures reached confluence, and microglia were isolated through mild trypsinization as previously described [47]. Briefly, cells were washed with pre-warmed DMEM-F12, 1% penicillin-streptomycin, and then treated with a 1:3 dilution of 0.25% trypsin in DMEM-F12 medium. Cells were incubated at 37 °C until the detachment of an intact layer of mixed glial cells, which left microglia firmly attached to the bottom of the well. Microglia rested for 48 h before performing any experiment.

#### 2.1.2. Polyphenols Screening In Vitro

Cortical microglia cultures were preincubated for 1 h with 1 of the 13 phenolic acid or polyphenol metabolites, separately, including ferulic acid (FA), vanillic acid (VA), 3′,4′-dihydrocaffeic acid (DHCA), epicatechin (EPI), catechin (CA), resveratrol (Resv), OMe-quercetin-glucuronide (3MQG), hippuric acid (HA), 3-(3′-hydroxyphenyl)propionic acid (3HPPA), 3-hydroxypropionic acid (3HPA), quercetin-glucuronide (QG), malvidin-3-*O*-glucoside (MG), and 3-hydroxybenzoic acid (3HBA) at a concentration of 10 μM (Table 1). The same volume of DMSO was added to the control samples. NLRP3 priming was induced by the addition of 400 ng/mL LPS for 3 h. Then, inflammasome activation was induced through the addition of 4.5 mM ATP for 45 min. Cell supernatants were collected for IL-1β determination and viability assessment. Experiments were run in triplicates (35).

#### 2.1.3. Determination of the MG Cytotoxicity

The cytotoxic effect of MG was assessed in primary microglia cultures. The cells were treated for 24 h with different concentrations of MG, ranging from 0.001 to 25 µM. The culture supernatants were taken, and the amount of LDH released was determined with the Cyto Tox 96^®^ NonRadioactive Cytotoxicity Assay (Cat#G1780, Promega, Madison, WI, USA), according to the manufacturer’s recommendations.

#### 2.1.4. Microglia Activation

The activation of cortical microglial murine cultures was induced differently depending on the target cytokine to be analyzed. First, microglia were treated with 10 µM MG for 1 h—or the same percentage of DMSO in the case of positive and negative controls. For TNF-α and IL-6 analysis, microglia were then treated with 400 ng/mL LPS for 6 h. For NLRP3 activation, microglia cultures were primed with 400 ng/mL LPS for 3 h prior to the addition of 4.5 mM ATP for 45 min. The cell supernatants were collected for cytokine analysis.

#### 2.1.5. Pyroptosis Assessment

The ability of different concentrations of MG to reduce pyroptosis was assessed by determining the LDH released by microglia cultures. For that purpose, microglia were incubated for 1 h with 0.001–25 µM MG, or DMSO as control, followed by addition of 400 ng/mL LPS for 3 h. Pyroptosis was then induced by the addition of 4.5 mM ATP. The LDH concentration in microglia supernatants was measured with the Cyto Tox 96^®^ NonRadioactive Cytotoxicity Assay (Cat#G1780, Promega, Madison, WI, USA), following the recommendations of the manufacturer.

#### 2.1.6. Dose–Response Curve for NLRP3 Inhibition

A dose–response curve was created to determine the potency of MG on NLRP3 inhibition. Microglia were incubated at 37 °C with increasing concentrations of MG—or the same DMSO percentage—ranging from 0.001 to 100 µM for 1 h. The transcriptional priming of the inflammasome components was induced by the addition of 400 ng/mL LPS for 3 h. NLRP3 activation was facilitated by adding 4.5 mM ATP for 45 min. Cell supernatants were taken for IL-1β analysis.

#### 2.1.7. Activation of the NLRC4 and AIM2 Inflammasomes

Microglia were pretreated with 10 µM MG for one hour, or with the same DMSO% in the case of positive and negative controls. NLRC4 activation was induced through the addition of 150 ng/mL LPS for 2 h, before transfecting 1 μg flagellin (2 h) with lipofectamine 2000 as a transfection reagent. For its part, the activation of the AIM2 inflammasome was produced by the addition of 100 ng/mL LPS for 2 h, followed by transfection of 1 μg poly (dA:dT) with lipofectamine 2000 for 2 h, according to the fabricator recommendations. For positive controls for inflammasome activation, the cells were treated with 150 ng/mL LPS prior to flagellin transfection—for the NLRC4 inflammasome—or with 100 ng/mL LPS before transfecting poly (dA:dT) in the case of the AIM2 inflammasome. For negative controls, cells were kept untreated after DMSO addition. After the different treatment paradigms, supernatants were collected for IL-1β measurement.

#### 2.1.8. Inhibition of the Caspase-1 Activity

The direct inhibition of the caspase-1 activity by 10 μM MG was assessed at the protein level, using purified and active caspase-1 (Cat#ab39901, Abcam, Cambridge, UK), and the fluorometric Caspase-1 Assay Kit (Abcam, Cat#ab39412, Abcam, Cambridge, UK), by following the protocol established by the fabricator.

#### 2.1.9. THP1 Macrophages

THP1 human monocytes were acquired from the American Type Culture Collection (ATCC^®^ TIB-202™). THP1 cells were cultured in RPMI-1640 medium, supplemented with 1% penicillin–streptomycin, 10% FBS, and 0.05 mM. Monocytes were differentiated into macrophages through treatment with 100 ng/mL phorbol 12-myristate 13-acetate (PMA) for 48 h. After that, THP1 macrophages were washed three times with culture medium (without FBS supplement) and let to rest for 24 h in complete medium. THP1 human macrophages were preincubated for 1 h with 10 µM MG or the same percentage of DMSO as control. Then, the transcriptional priming of the different components of the NLRP3 inflammasome pathway (*Nlrp3*, *Il-6*, *Il-8*, *Tnf-α*, *Il-1β*, and *Caspase-1)* was induced through the addition of 400 ng/mL LPS. Cells were incubated at 37 °C for 3 h and then harvested for RNA extraction and gene expression assessments [35].

### 2.2. In Vivo Studies

#### 2.2.1. Animals

Wild type C57BL/6J male mice were obtained from the Jackson Laboratory (JAX #000664). Animals were 8 weeks old at the beginning of the experiment and were maintained on a 12/12 h light/dark cycle in a temperature-controlled (20 ± 2 °C) vivarium. Food and water were available *ad libitum*. All animal experiments complied with the guidelines approved by the Institutional Animal Care and Use Committee of the Icahn School of Medicine at Mount Sinai.

#### 2.2.2. Malvidin-3-*O*-Glucoside (MG) Treatment

MG was purchased from Millipore Sigma (Cat#PHL89728; Burlington, MA, USA) and Extrasynthese (Cat#0911 S; Rhone, France). Purity (HPLC) was ≥95%. It has been shown to be stable when stored at < 15 °C, in a dry and dark place. According to the Company, MG is purified from a natural source. The anthocyanin was diluted in drinking water to the desired concentration of 12.5 mg/kg BW and stored at 4 °C until further use. The dose was selected taking into account the nature of the compound, equivalent doses, length of treatment, animal strain, and way of administration that have been used in the published literature [41,48,49,50]. Treatments were administered via intragastric gavage daily for 6 weeks (42 days). A 2-week pre-treatment was followed before CUS studies. On the last day of treatment, blood was collected by heart puncture as well as brain tissue from the prefrontal cortex, hippocampus, and cortex for further analysis.

MG bioavailability was examined by pharmacokinetic (PK) studies using plasma and brain tissue samples from mice administered via intragastric gavage at a dose of 12.5 mg/kg BW/day for 7 days. The samples were stored at −80 °C.

#### 2.2.3. Behavioral Studies

##### Chronic Unpredictable Stress (CUS) Protocol

The effect of MG on depressive and anxiety-like behavior was studied in a model of unpredictable stress (CUS). Animals were randomly assigned to three different groups (*n* = 7–10), including vehicle, those receiving regular drinking water, vehicle + CUS, those receiving drinking water and subjected to stress, and MG, those animals receiving MG treatment dissolved in drinking water. A pre-treatment was followed two weeks prior to the initiation of the stress protocol. Mice were then subjected to CUS for 28 days which consisted of two stressors per day separated by at least 4 h. Stressors followed included 45 °C cage tilt for 17 h, wet bedding for 7 h, no bedding for 17 h, food and/or water deprivation for 17 and 7 h, respectively, crowding for 1 h, restrain or restrain with predator odor for 1 h, hot drier alternating cold and hot temperature for 10 min, lights on, foot shock (2 s, 0.8 mA), cold swim for 5 min, tail suspension for 1 h, cage shake for 1 h. Sometimes a combination of two stressors was used at the same time (Table 2). Food and water consumption were monitored weekly while weights were recorded daily, and no significant changes were observed for either food or water consumption throughout the testing period. Animals were weighed daily. Significant changes were observed at different time points after stress (day 15, 24, 29, 32, 35, and 40), as expected. However, they recovered fast, returning to their regular weight the day after (Appendix A). Animals were sacrificed immediately after the last day of treatment and brain tissue was collected.

##### Elevated Plus Maze (EPM)

All behavioral assays were conducted 3 h after the beginning of the light cycle. EPM was performed on days 1, 15, 28, and 42 (EPM1, EPM2, EPM3, and EPM4, respectively) for the assessment of anxiety-related behavior. The EPM apparatus consisted of a “+”–shaped maze elevated above the floor with four arms consisting of two exposed arms also known as “open arms” and two arms enclosed in opaque gray plastic walls 80 cm long and 5 cm wide known as “closed arms”. Each arm of the maze is attached to 50 cm high stainless–steel legs. The maze was situated on a flat and stable base so that there were similar levels of illumination on both open and closed arms. The animals were placed in the center or the neutral area facing the open arms and allowed to explore freely the apparatus for 5 min. Afterwards, the mice were returned to their cage. All test trials were automatically recorded by a video-tracking system mounted above the maze and analyzed with ANY–Maze tracking software (v 6.0; Stoelting, Wood Dale, IL, USA). In order to minimize noise and avoid erratic animal behavior, the data collection was performed in a separate room from where the maze was located.

##### Forced Swim Test (FST)

The forced swim test was performed on the last day of the experiment (day 43) for the study of depression-like behavior. The test consists of placing the animals in 4 L Pyrex glass beakers with a 16 cm diameter containing 2 L of water at a temperature of 24 ± 1 °C. Two side-facing cameras under ambient light conditions were used to monitor the animals’ behavior for 6 min, with the first minute being excluded from the analysis.

#### 2.2.4. MG bioavailability: Pharmacokinetics

MG was administered by oral gavage to mice at a dose of 12.5 mg/kg BW/day for 7 days. On day 7, animals were given their last dose and sacrificed at 0 h, 15 m, 30 m, 1 h, 2 h, 8 h, and 24 h post-treatment. Animals were anesthetized with isoflurane and blood was collected by cardiac puncture into heparinized tubes. Blood was spun down at 4 °C at 4000× *g* for 15 min for plasma collection. At a 1:10 dilution, 2% formic acid was added to plasma. After blood collection, animals were transcardially perfused with 10 mL cold PBS, and brains were extracted. Each half was weighed and a 3× volume of 0.2% formic acid was added to each sample before homogenization with a bead homogenizer. All samples were snap-frozen in dry ice and stored at −80 °C until shipment. Samples were analyzed by the Department of Plant Biology, SEBS, at Rutgers University (New Brunswick, NJ, USA).

##### Sample Preparation

Plasma and brain tissue samples were thawed on ice and then processed at room temperature (RT). For analyte extraction from biomatrices, an aliquot of 200 μL plasma was thawed on ice followed by adding 50 μL of each IS solution (ca. 0.2 μg/mL quercetin-glucoside). The analytes were then extracted with 500 μL of ethyl acetate, vigorously vortexed for 10 s, sonicated in ice water for 10 min, and then centrifuged at 5000× *g* for 5 min. The supernatant was collected in an Eppendorf tube containing 20 μL 2% ascorbic acid methanol solution. The precipitate was then extracted in a similar manner two more times. The pooled supernatant was dried using a speed vacuum. The residue was reconstituted in 150 μL of 60% methanol containing 0.1% formic acid, centrifuged at 16,000× *g* for 10 min before Liquid Chromatography-Mass Spectrometry (LC-MS) analysis. The brain samples were processed in a similar procedure as plasma, except the following: the processed tissue amount used was 500 μL and 100 μL of 4% HCl was added before extraction to denature and precipitate proteins.

##### LC–MS Method

For each sample extract, 2.5 μL was injected into an Agilent UPLC-QqQ/MS system in the first run, and 10 μL was injected in the second run for analysis under dynamic multiple reaction monitoring (dMRM) mode. Because malvidin-glucuronide, malvidin sulfate, and me-malvidin-glucoside have no reference standards, we referred to earlier publications for their MRM transitions and MRM parameters. The concentrations were calculated based on the correction factor of MW ratio of malvidin-3-glucoside.

#### 2.2.5. Quantitative PCR

Total RNA was extracted from brain tissue samples using TRIzol (Cat#15596026, Invitrogen, Waltham, MA, USA) followed by the RNeasy Mini Kit (Cat#74104, Qiagen, Hilden, Germany) according to the manufacturer’s instructions. RNA was reverse transcribed using the High–Capacity cDNA Reverse Transcription Kit (Cat#4368813, ThermoFisher, Waltham, MA, USA) according to the manufacturer’s instructions. Expression of IL–1β (forward: 5′TTCAGGCAGGCAGTATCACTC3′; reverse: 3′ CCACGGGAAAGACACAGGTAG5′) was measured with PowerUp SYBR Green Master Mix (Cat#A25742, ThermoFisher) in an ABI PRISM 7900HT Sequence Detection System at the Icahn School of Medicine qPCR CoRE. Data were normalized to hypoxanthine phosphoribosyltransferase (*Hprt*) internal control using the 2^−ΔΔCt^ method (forward: 5′CCCCAAAATGGAGACCAGAC3′; reverse: 3′ AACAAAGTCTGGCCTGTATCC5′).

### 2.3. ELISA Assay

Microglia cell culture supernatants were centrifuged (500 rpm) for 5 min at 4 °C and stored at −80 °C until further use. The concentration for IL-1β, IL-6, and TNF-α was determined using the corresponding ELISA kits (Cat#DY401, #DY410, #DY406, R&D, Minneapolis, MN, USA), following the protocols designed by the manufacturers.

Brains were rinsed in cold PBS and the prefrontal cortex and hippocampus were dissected out and immediately frozen in dry ice. For protein analysis, brain tissue was lysed in 1× Cell Lysis Buffer (Cat#9803, Cell Signaling, Danvers, MA, USA) supplemented with 1 mM PMSF (Cat#8553, Cell Signaling, Danvers, MA, USA) and a protease inhibitor cocktail (Cat#11836170001, Sigma-Aldrich, St. Louis, MO, USA) using a bead homogenizer. Protein concentration was measured using a Pierce BCA kit (Thermo Scientific #23225). Samples were stored at −80 °C until further analysis. IL-1β levels were quantified via ELISA (Cat#DY401, R&D, Minneapolis, MN, USA), according to the manufacturer’s instructions.

### 2.4. Statistical Analysis

All values are expressed as the mean ± standard error of the mean. Scatter plots represent individual values depicted as black dots. One-way ANOVA with Tukey’s post-hoc was performed for overall comparisons. One-way ANOVA with Dunnett’s multiple comparisons test was performed to compare the control group to the test groups. A *t*-test analysis was used to compare two experimental groups. Outliers (2 SD from the mean) were removed from the analysis. All statistical analyses were performed using GraphPad Prism (version 9.1.1; San Diego, CA, USA). Significant differences were set to * *p* ≤ 0.05, ** *p* ≤ 0.01, *** *p* ≤ 0.001, **** *p <* 0.0001. No significant differences will be shown as ns.

## 3. Results

### 3.1. The Natural Polyphenol MG Reduces NLRP3–Mediated IL–1β Production in Murine Primary Cortical Microglia

To determine the anti-inflammatory potential and, more importantly, to identify NLRP3 inflammasome inhibitors, various phenolic compounds were selected, including ferulic acid (FA), vanillic acid (VA), 3′,4′-dihydrocaffeic acid (DHCA), epicatechin (EPI), catechin (CA), resveratrol (Resv), OMe-quercetin-glucuronide (3MQG), hippuric acid (HA), 3-(3′-hydroxyphenyl)propionic acid (3HPPA), 3-hydroxypropionic acid (3HPA), quercetin-glucuronide (QG), malvidin-3-O-glucoside (MG), and 3-hydroxybenzoic acid (3HBA) (Figure 1). We performed a cell-based screening, where we pretreated murine primary cortical microglia cells with 10 μM of the individual metabolites before the induction of priming and activation with LPS and ATP, respectively, leading to IL-1β release in cell supernatants. Among the 13 assayed metabolites, the MG compound is the most striking treatment able to inhibit the NLRP3-mediated secretion of IL-1β levels significantly at the assessed concentration compared to control (Ctrl) microglial cells (pre-treated with DMSO% and LPS + ATP, only) (*** *p* = 0.0004). Other natural polyphenols, Resv, 3MQG, and HA (** *p* = 0.0012, * *p* = 0.0182, * *p* = 0.0160, respectively), were also shown to provide a protective anti-inflammatory effect (Figure 1). This reveals the higher inhibitory activity of MG against NLRP3-mediated IL-1β release when compared with the rest of screened metabolites consequently selected for further characterization.

### 3.2. MG Is Not Cytotoxic and Elicits Anti–Pyroptotic Properties in Murine Primary Cortical Microglia

To further characterize the use of MG as an anti-inflammatory agent, the potential toxicity was determined. MG treatment did not show cytotoxicity using different concentrations (0.001, 0.01, 0.1, 1, 5, 25 μM), as shown by measuring the LDH release to cell supernatant (Figure 2A). To investigate the ability of MG to inhibit NLRP3-mediated canonical pyroptosis, we measured the LDH release. Microglia were treated with different concentrations of MG (0.001, 0.01, 0.1, 1, 5, 25 μM) prior to NLRP3 inflammasome induction with LPS and ATP. A significant reduction of LDH formation was observed from 0.1 μM to the highest concentration of 25 μM (* *p* = 0.0107, *** *p* = 0.0002, ** *p* = 0.001, and *** *p* = 0.0008, respectively) (Figure 2B). In summary, these results demonstrate the potential anti-inflammatory and anti-pyroptotic properties exerted by MG in primary cortical microglia, besides being a non-cytotoxic polyphenol, showing the importance of MG in the canonical pyroptosis mechanisms mediated by inflammasome formation and therefore in the development of innate immune responses.

### 3.3. MG Potently Inhibits NLRP3, NLRC4, and AIM2 Inflammasome Assembly

Inflammasome assembly mediates the canonical pyroptotic pro-inflammatory form of cell death [51]. Hence, we further analyzed the inhibitory effect of MG on the production of IL-1β upon activation of NLRP3, NLRC4, and AIM2 inflammasomes in primary microglial cells. The NLRP3 inflammasome was inhibited in a dose-dependent manner as shown by the decrease in IL-1β levels, being significantly reduced at 1, 5, 25, and 100 µM compared to non-treated cells (*** *p* = 0.0009, **** *p <* 0.0001, **** *p <* 0.0001, and **** *p <* 0.0001, respectively) (Figure 3A). The nonlinear fit of the data to a dose-response inhibition model allows the calculation of an IC50 value of 0.94 µM (Figure 3B). We then investigated the ability of MG to inhibit other IL-1β-producing inflammasomes, including NLRC4 and AIM2. For that purpose, we measured IL-1β produced by primary microglia cultures upon stimulation with specific activators of both inflammasomes, in the presence (1 and 10 µM) and absence of MG. As shown in Figure 3C,D, treatments at 1 and 10 µM MG inhibit both NLRC4 (** *p* = 0.0094, **** *p <* 0.0001, respectively) and AIM2 (** *p* = 0.0088, **** *p <* 0.0001, respectively). The findings of these studies show the direct effect of MG on inflammasome formation. Furthermore, MG might act at the common priming stage of the three inflammasomes by reducing the expression of the ASC adaptor protein, caspase-1, and/or IL-1β, or it might act at some step located downstream of inflammasome activation, such as ASC recruitment and oligomerization or caspase-1 activity. Therefore, with the objective of providing a description of the MG-specific target, we studied its effect on the aforementioned processes.

### 3.4. MG Does Not Interfere with The transcriptional Priming of Inflammasome Components

We next investigated whether MG inhibits the inflammasome pathway by reducing the expression of the different proteins involved in the pathway that leads to IL-1β release. As shown in Figure 4, the treatment of human macrophages with 450 ng/mL LPS significantly increases the expression of the NLRP3 inflammasome pathway components IL-6, TNF-α, and caspase-1 compared to the LPS untreated cells (*** *p* = 0.0008, **** *p* < 0.0001, and **** *p* < 0.0001, respectively) (Figure 4A–C). However, MG does not have any effect on the IL-6 and TNF-α amounts produced by microglia cultures upon stimulation with LPS, which points toward the specificity of the phenolic metabolite for a stage of the inflammasome pathway located after the activation of the NLR receptor and the release of IL-1β (ns: *p* = 0.089; ns: *p* = 0.9980, respectively) (Figure 4A,B). As shown in Figure 4C, the activity of caspase-1 at the protein level ameliorates after MG treatment at 10 µM compared to those untreated microglial cells (**** *p* < 0.0001). Moreover, the pretreatment with 10 µM MG does not affect the mRNA levels of any of the following genes: *Nlrp3*, *Il*-*6*, *Il*-*8*, *Tnf*-*α*, *Il*-*1β,* and *Caspase*-*1* (Appendix A), indicating that the polyphenol works downstream of inflammasome activation rather than at the transcriptional level. These findings suggest mechanistic action downstream of initial microglial activation through reduction of caspase-1.

### 3.5. MG Treatment Decreases Anxiety and Depressive-Related Behavior in a Mouse Model of Chronic Unpredictable Stress (CUS)

Previous studies in our lab have reported the beneficial effect of polyphenolic components against stress-induced depressive and anxiety-like behaviors in a CUS mouse model, as well as the effect on pro-inflammatory mediators in the brain [52]. Our goal here is to verify the individual bioactivity of MG alone for the treatment of chronic stress-induced depression and anxiety. Chronic unpredictable stress was subjected to the mice for 28 days, consisting of a plethora of different stressors (Figure 5) (Table 2), and the effect of MG on the resulting stress-induced anxiety was evaluated using the elevated plus maze. To monitor anxiety progress, this assay was performed at four time points throughout the experiment: EMP1: before MG pre-treatment, EPM2: one day before the start of CUS, EPM3: 15 days after the start of CUS, and EPM4: one day after finishing CUS. No significant differences were observed among the three experimental groups for each independent EPM, neither for distance traveled nor for time spent in closed or open arms (ns: *p* > 0.05) (Figure 6A-C) (Appendix A). Most importantly, oral gavage administration alone does not provide a significant stressful effect in the animals, as we did not observe significant differences between EPM2 and EPM1. However, when we compare EPM4 to EPM2, that is, after and before CUS, stressed non-treated animals (Vehicle + CUS) had a stronger preference for closed arms over open arms with respect to vehicle non-stressed mice (** *p* = 0.0037) (Figure 6D,E). MG-treated animals exhibited less anxiety, as they were more prone to exploring the open field compared to those non-treated animals subjected to CUS (closed arms: ** *p* = 0.0082; open arms: * *p* = 0.0353) (Figure 6D,E).

Furthermore, we further explored depression-like behavior using the forced swim test. We observed an increase in immobilization in non-treated stressed animals compared to vehicle non-stressed (* *p* = 0.0463) (Figure 6F). MG treatment ameliorated motility, reversing the CUS-induced depression-like behavior (* *p* = 0.0378). Together, these data show that MG can improve the CUS-induced anxiety and depression-related behavior as indicated by more exploration of open spaces and an increase in motility, therefore promoting resilience in those animals subjected to chronic stress.

### 3.6. Il-1β Is Downregulated after MG Treatment in a Mouse Model of Chronic Unpredictable Stress (CUS)

IL-1β cleavage and activation occur after inflammasome assembly and activation as a response to chronic stress exposure. Therefore, we next evaluated whether MG was able to act in downstream signaling after inflammasome induction in our in vivo model of CUS. Molecular studies using RT qPCR show that there is a downregulation of *Il*-*1β* expression in the hippocampus in those CUS-induced stressed animals treated with MG compared to those untreated and not subjected to stress (* *p* = 0.0398) (Appendix A). However, there is no effect of MG in the prefrontal cortex (ns, *p* = 0.6026) (Appendix A). The protein levels analyzed by ELISA show a decrease in the hippocampus and prefrontal cortex for those stressed animals treated with MG (** *p* = 0.0038, ** *p* = 0.0022) (Figure 7A,B, respectively). The in vivo studies show the ability of MG to halt the release of IL-1β in the hippocampus, improving the CUS-induced inflammation, suggesting that this natural compound plays a role both at the transcriptional level and downstream of inflammasome activation. However, MG displays a differential biological effect in the prefrontal cortex, where there is an effect only at the protein level.

### 3.7. MG Is Rapidly Degraded in Brain and Plasma after Intragastric Administration: Pharmacokinetics Study

After we identified the therapeutic actions of MG on the treatment of inflammasome-induced inflammation, both in vitro and in vivo, bioavailability studies were performed to determine to what extent this polyphenol was present in the systemic circulation and brain. We administered MG orally by intragastric gavage, at a dose of 12.5 mg/kg BW, daily for 7 days for assessment by pharmacokinetic studies and analysis by the UPLC-QqQ-MS/MS method. Only the two metabolites malvidin-3-glucoside and malvidin glucuronide were shown in the brain and plasma (Figure 8). Malvidin sulfate and me-malvidin-glucoside were not detected. In the brain, MG’s peak is set at 30 min (0.08 nM) when levels start to gradually decrease. However, malvidin glucuronide was found to be present in higher concentrations than the other metabolite, reaching the peak at 15 min (0.17 nM) and decreasing gradually after this time point (Figure 8A). In plasma, MG is metabolized in a fast-paced way, to the point that it reaches the peak concentration (20.93 nM) after 15 min, when the levels start to decrease abruptly and continue to reduce. Malvidin glucuronide, on the other hand, does not reach more than 1.5 nM, maintaining levels rather low but stable until 24 h after administration (Figure 8B). Therefore, from the administration of MG at a dose of 12.5 mg/kg BW administered by gastric gavage, daily for 7 days, only two metabolites were detected in plasma and in the brain. MG is seen to be rapidly degraded through hepatic metabolism via glucuronidation, and also has low protein binding quickly reaching therapeutic concentrations in peripheral tissue including the brain.

## 4. Discussion

The importance of psychosocial stress is of notice when it is considered a risk factor that accounts for 75–90% of diversified stress-related diseases [3]. Chronic psychological stress can induce a persistent immune reaction leading to an increase in the number of proinflammatory cytokines contributing to a cognitive impairment such as depression [4,15]. Inflammasomes are protein complexes that represent the first mechanism of defense in the innate immune system against danger signals of diverse natures, as they also are key factors in the pathology of inflammation-related illnesses [53,54]. Consequently, they provide an interesting profile as molecular targets to address inflammasome-mediated inflammation and the resulting behavior impairment. As a therapeutic strategy in our study, MG was selected among 13 natural polyphenols in a cell-based screening as the most potent agent to prevent the NLRP3-mediated secretion of IL-1β. Moreover, MG was seen to be non-cytotoxic at any of the concentrations used, ranging from 0.001 to 25 μM, making this agent a more plausible anti-inflammatory agent for clinical purposes. Inflammasomes engage in a pro-inflammatory form of cell death or pyroptosis, and this study shows the inhibitory properties of MG in the canonical pyroptosis mechanisms mediated by inflammasome formation and therefore in the development of innate immune responses. These results are in accordance with Esposito et al. [45], who show a reduction in the expression of the pro-inflammatory genes *Cox*-*2*, *iNOS,* and *Il*-*1β* when using malvidin-3-glucoside from wild blueberries in LPS-treated macrophages. Additionally, it also exhibits anti-cytotoxicity properties against human monocytic leukemia cells and HT-29 colon cancer cells [54,55]. Likewise, MG has been used in a number of in vivo studies in order to investigate its potential anti-inflammatory activity in TNF-α-induced inflammatory responses [43].

Chronic stress-related disorders may induce an accumulation of cellular damage that will eventually contribute to pathogenic immune responses. The NLR family responds to the resulting cellular disturbances by oligomerization, inflammasome formation, and activation [56,57]. Inflammasomes will serve as essential factors in the response of the innate immune system following exogenous or endogenous stimuli and also mediate canonical pyroptosis [31]. In our studies, we show the effect of MG to target specifically the NLRP3, NLRC4, and AIM2 inflammasome complexes in a dose-dependent manner proving the potential role of this polyphenol in the development of innate immune responses following stress. The aforementioned data are in agreement with other in vivo studies using different polyphenol extracts to attenuate NLRP3 activation in obesity-induced low-grade inflammation and LPS-induced liver injury [58,59]. Along with this data, we further examined the beneficial impact of MG on the canonical inflammatory downstream signaling, indicating the inhibitory properties on levels of caspase-1 activity at the protein level but no effect on IL-6 or TNF-α. However, MG had no impact on transcriptional priming of the inflammasome components as observed on the mRNA expression of *Nlrp3*, *Il*-*6*, *Il*-*8*, *Tnf*-*α*, *Il*-*1β,* and *Caspase*-*1*, suggesting that MG might act downstream of microglial activation. These findings show that the polyphenol works downstream of inflammasome activation rather than at the transcriptional level, suggesting a mechanistic role downstream of initial microglial activation, by reducing the levels of caspase-1.

Inflammasome activation acts as a pivotal bridge through which psychological and physical stressors can contribute to the development of psychological impairment [60]. Zhang et al. [61] showed that mice subjected to chronic mild stress displayed an increase in IL-1β, caspase-1, NLRP3, and ASC in the hippocampus. The inhibition of NLRP3 led to diminishing central and peripheral inflammation. There are a number of studies using a combination of polyphenol extracts, collected from natural sources, suggesting a therapeutic potential to alleviate or arrest sleep deprivation or stress-dependent cognitive deterioration [62,63]. In addition, the beneficial effect of polyphenolic components against AD-type cognitive decline, Aβ peptide accumulation in the brain, and tauopathies has been reported [64,65,66,67]. Accordingly, our data showed that MG improved CUS-induced anxiety and depression-related behavior, as indicated by more exposure to open spaces in the EPM and an increase in motility in the forced swim test, therefore promoting resilience in those animals subjected to chronic stress.

Moreover, MG exerted an anti-inflammatory effect in the hippocampus as seen by the downregulation of *Il*-*1β* gene expression as well as IL-1β protein levels in mice exposed to CUS. However, MG had no impact on gene regulation besides at a protein level only in the prefrontal cortex. This drives us to the idea that MG might exhibit a different biological function depending on the distinct inflammatory phenotype-associated tissue. Further studies to unravel the brain tissue-specific pathways leading to a specific inflammatory phenotype should be considered. These results are in line with the research by Westfall et al. [52], where treatment of mixed polyphenol metabolites or symbiotic metabolites enhanced the stress-induced depressive and anxiety-like behaviors in a CUS mouse model, as well as the effect on pro-inflammatory mediators in the brain.

In our studies, MG was administrated by intragastric gavage at a dose of 12.5 mg/kg BW, daily for 7 days. Malvidin-3-glucoside and malvidin glucuronide, two bioactive polyphenols, were found in the brain and plasma, as expected for in vivo PK studies. The precursor malvidin–glucoside is first degraded to aglycone and then forms glucuronide conjugates. It is also known that only 5–10% of orally consumed polyphenols is absorbed in the small intestine, while 90–95% is further broken down by gut microbiota fermentation in the colon or large intestine [68,69]. Frolinger et al. [70] showed that polyphenol bioavailability and biological function depend on gut microbiota composition in order to provide memory improvement in a model of sleep deprivation. Furthermore, some studies state the capacity of the resulting low-molecular-weight phenolic compounds, absorbed more efficiently by the gastrointestinal epithelium, to be physiologically relevant at low concentrations of the order of µM to sub-µM [71,72,73]. According to Cruz et al. [74], the aglycone forms conjugated with glucuronic groups are more relevant in vivo. Hence, despite the fact that malvidin glucoside is seen to be rapidly degraded, the levels either of the compound itself or the resulting metabolites are enough to be physiologically effective and exert anti-inflammatory actions in our model of chronic stress.

## 5. Conclusions

In conclusion, these data contribute to the better understanding of the anti–inflammatory role that MG plays on inflammasome-associated inflammation. More importantly, this study depicts the therapeutic potential for a single polyphenol, MG, to prevent deregulation of innate immunity and aid in the treatment of anxiety and depressive-related behavior due to chronic unpredictable stress.

## Figures and Tables

**Figure 1 biomedicines-10-01264-f001:**
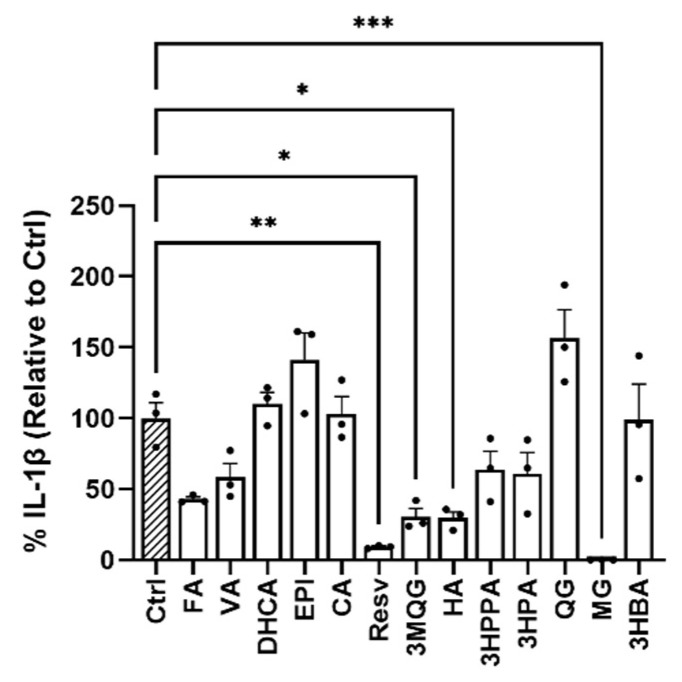
In-vitro screening of polyphenols including ferulic acid (FA), vanillic acid (VA), 3′,4′-dihydrocaffeic acid (DHCA), epicatechin (EPI), catechin (CAT), resveratrol (Resv), OMe-quercetin-glucuronide (3MQG), hippuric acid (HA), 3-(3′-hydroxyphenyl) propionic acid (3HPAA), 3-Hydroxypropionic acid (3HPA), quercetin-glucuronide (QG), malvidin-3-*O*-glucoside (MG), and 3-hydroxybenzoic acid (3HBA). The ability of 10 μM of the natural compounds to inhibit NLRP3-mediated IL-1β production was tested using murine microglia primary cultures primed and activated with LPS and ATP, respectively. The IL-1β levels in culture supernatants were determined by ELISA. Experiments were carried out in triplicate, and samples were in technical duplicates. Statistical significances are referred to control samples (cells pre-treated with the same DMSO% and LPS + ATP). All graphs represent mean ± s.e.m. Scatter plots represent individual values depicted as black dots. Significance levels, as calculated by one-way analysis of variance, are indicated as: Ctrl vs. Resv: ** *p* = 0.0012, Ctrl vs. 3MQG: * *p* = 0.0182; Ctrl vs. HA: * *p* = 0.0160; Ctrl vs. MG: *** *p* = 0.0004.

**Figure 2 biomedicines-10-01264-f002:**
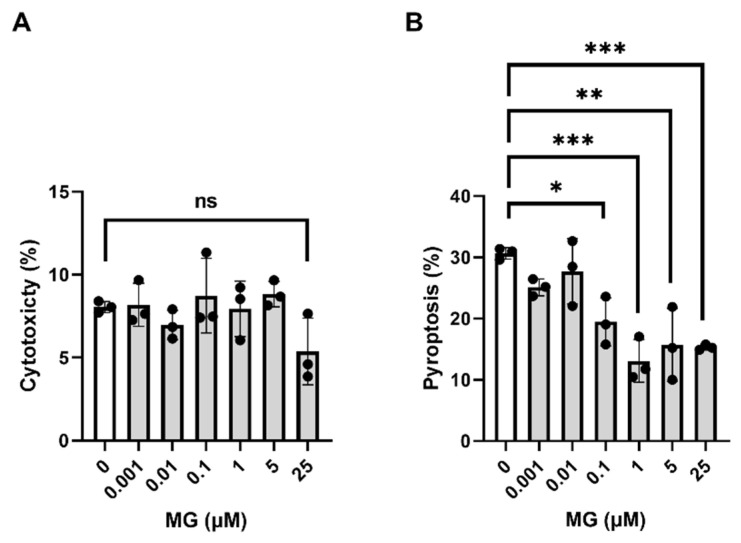
(**A**) Cytotoxicity exerted by increasing concentrations of MG. The LDH released by primary microglia cultures treated with MG concentrations ranging from 0.001 to 25 μM was determined. (**B**) Antipyroptotic effect of MG. Primary microglia cultures were pre-treated with DMSO (control) or increasing concentrations of MG prior to pyroptosis induction with LPS and ATP. Pyroptosis was determined by measuring cell death using an LDH assay. Experiments were run in triplicate, and samples were in technical duplicates. Statistical significances are referred to control samples (cells pre-treated with the same DMSO% for panel (**A**) and the same DMSO% and LPS + ATP for panel (**B**). All graphs represent mean ± s.e.m. Scatter plots represent individual values depicted as black dots. Significance levels, as calculated by one-way analysis of variance, are indicated as: (**A**) 0 vs. 25 µM: ns (no significant), *p* = 0.1702; (**B**) 0 µM vs. 0.1 µM: * *p* = 0.0107; 0 µM vs. 1 µM: *** *p* = 0.0002; 0 µM vs. 5 µM: ** *p* = 0.001; 0 µM vs. 25 µM: *** *p* = 0.0008.

**Figure 3 biomedicines-10-01264-f003:**
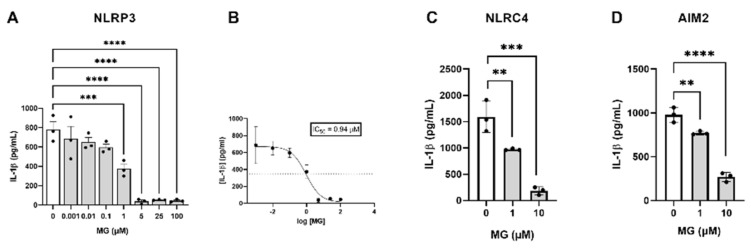
(**A**) Dose-dependent inhibitory effect of MG. (**B**) Dose-response curves for MG. The IC50 value was obtained by nonlinear regression of the data. In both (**A**) and (**B**), primary murine microglia cultures were treated with increased MG concentrations prior to NLRP3 activation with LPS and ATP. (**C**) Inhibitory effect of MG on the NLRC4 inflammasome. Primary microglia cells were pre-treated with MG, and then NLRC4 specific activation was induced with LPS followed by treatment with flagellin. (**D**) Inhibition of the AIM-2 inflammasome by MG. Murine primary microglia were treated with MG before AIM-2 priming and activation with LPS and poly (dA:dT). In all panels, the inhibition of inflammasome pathways by MG was determined by measuring the IL-1β concentration in culture supernatants through ELISA. All the experiments were conducted in triplicate, and samples were run in technical replicates. Statistical significances are referred to control samples (cells pre-treated with the same DMSO% and concentration of the corresponding inflammasome activators). All graphs represent mean ± s.e.m. Scatter plots represent individual values depicted as black dots. Significance levels, as calculated by one-way analysis of variance, are indicated as (**A**) 0 µM vs. 1 µM: *** *p* = 0.0999; 0 µM vs. 5 µM: **** *p <* 0.0001; 0 µM vs. 25 µM: **** *p <* 0.0001; 0 µM vs. 100 µM: **** *p <* 0.0001. (**C**) 0 µM vs. 1 µM: ** *p* = 0.0094; 0 µM vs. 10 µM: *** *p* = 0.001; (**D**) 0 µM vs. 1 µM: ** *p* = 0.0088; 0 µM vs. 10 µM: **** *p <* 0.0001.

**Figure 4 biomedicines-10-01264-f004:**
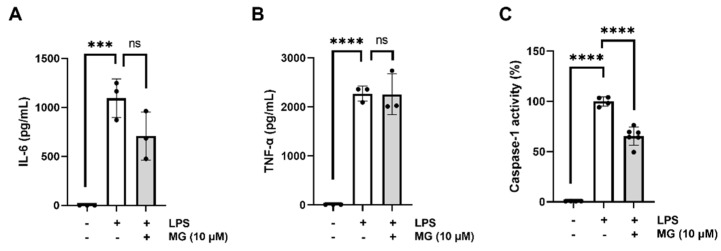
(**A**,**B**) Effect of MG on IL--6 and TNF-α production. Microglia cultures were pre-treated with MG before inducing IL-6 and TNF-α release to cell supernatants with LPS. (**C**) Inhibition of caspase-1 activity. The inhibition of the caspase-1 activity at the protein level was measured by determining the activity of purified caspase-1 in the presence (+) and absence (−) of MG (10 µM) treatment through a fluorimetric assay. Statistical significance is referred to control samples (cells pre-treated with the same DMSO% alone (−) and LPS (+)). All graphs represent mean ± s.e.m. Scatter plots represent individual values depicted as black dots. Significance levels, as calculated by one-way analysis of variance, are indicated as: (**A**) LPS untreated vs. 0 µM MG: *** *p* = 0.008; 0 µM MG vs. 10 µM MG: ns (no significant), *p* = 0.089; (**B**) LPS untreated vs. 0 µM MG: **** *p <* 0.0001; 0 µM MG vs. 10 µM MG: ns (no significant), *p* = 0.998; (**C**) LPS untreated vs. 0 µM MG: **** *p <* 0.0001; 0 µM MG vs. 10 µM MG: **** *p <* 0.0001.

**Figure 5 biomedicines-10-01264-f005:**
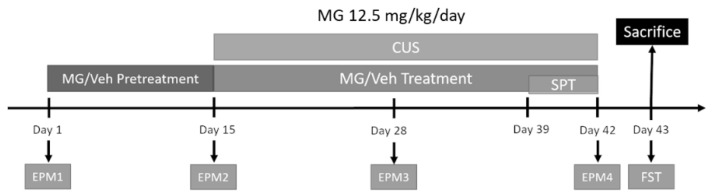
Schedule for malvidin-3-*O*-glucoside (MG) treatment and chronic unpredictable stress (CUS) protocol. CUS was subjected to the mice for 28 days, consisting of a plethora of different stressors, and the effect of MG on the resulting stress-induced anxiety was evaluated using the elevated plus maze (EPM). To monitor anxiety progress, this assay was performed at four time points throughout the experiment: EMP1: before MG pre-treatment, EPM2: one day before starting CUS, EPM3: 15 days after starting CUS, and EPM4: one day after finishing CUS.

**Figure 6 biomedicines-10-01264-f006:**
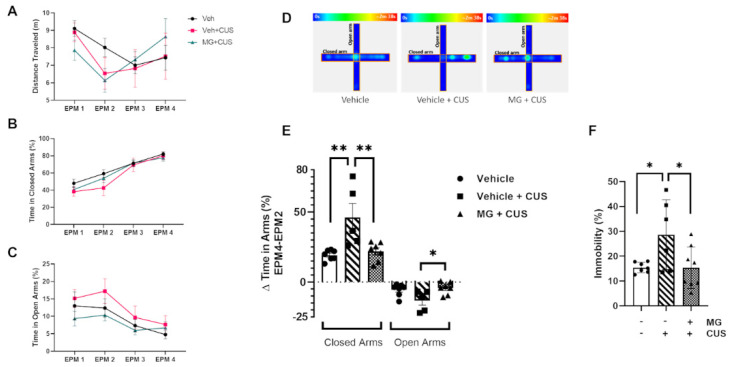
EPM trends are shown for distance traveled (m) (**A**), time in closed arms (%) (**B**), and time in open arms (**C**) throughout the experiment. No significant differences were observed. All graphs represent mean ± s.e.m. Heat map representing the time spent in closed and open arms in the Y-maze (**D**). Difference between the two time points after and before CUS: EPM4-EPM2 (**E**). Forced swim test (FST) (**F**), as indicator of depressive-related behavior, in the presence (+) and absence (−) of MG (10 µM) treatment, and in the presence (+) and absence (−) of CUS. All graphs represent mean ± s.e.m. Significance levels, as calculated by one-way analysis of variance, are indicated as: (**E**) for closed arms: ● vs. ■ ** *p* = 0.037; ■ vs. ▲** *p* = 0.0082; for open arms: ■ vs. ▲* *p* = 0.0353; (**F**) ●vs. ■* *p* = 0.0463; ■ vs. ▲* *p* = 0.0378.

**Figure 7 biomedicines-10-01264-f007:**
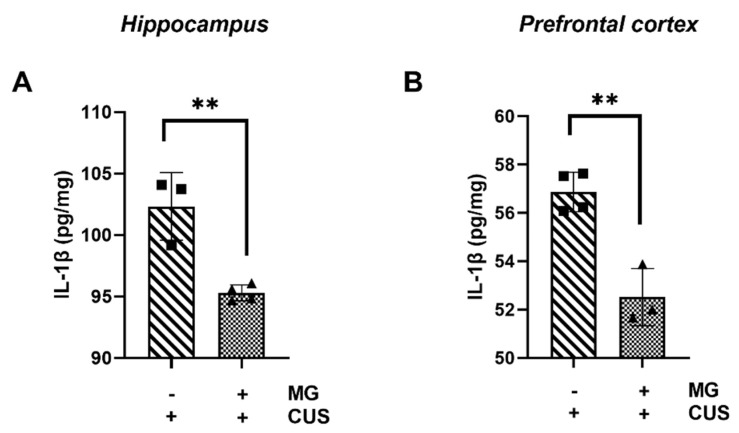
ELISA analysis for IL-1β protein levels in the hippocampus (**A**) and in the prefrontal cortex (**B**). All graphs represent mean ± s.e.m. Significance levels, as calculated by unpaired *t* test, are indicated as: (**A**) ■ vs. ▲** *p* = 0.0038; (**B**) ■ vs. ▲** *p* = 0.0022.

**Figure 8 biomedicines-10-01264-f008:**
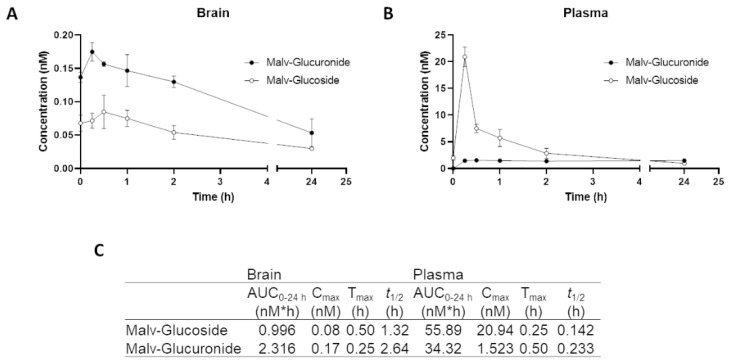
Pharmacokinetic study of malvidin glucoside administered by oral gavage daily at a concentration of 12.5 mg/kg for 7 days. MG levels in brain (**A**) and plasma (**B**) were measured after 30 min, 1, 2, 8, and 24 h after last administration (*n* = 3 for each experimental group). Only the metabolites malvidin glucuronide and malvidin glucoside were detected. (**C**) Average pharmacokinetic parameters in brain and plasma following 7 days of MG treatment.

**Table 1 biomedicines-10-01264-t001:** Information of polyphenols, including phenolic acids and polyphenol metabolites.

Name	Nature	Company	Purity
Ferulic acid (FA)	Phenolic acid	MP Biochemicals (Irvine, CA, USA)	No information
Vanillic acid (VA)	Phenolic acid	Sigma Aldrich (St. Louis, MO, USA)	(HPLC) ≥ 97%
3′,4′-dihydrocaffeic acid (DHCA)	Phenolic acid	Sigma Aldrich (St. Louis, MO, USA)	(titration by NaOH) 97.5-102.5%
Epicatechin (EPI)	Polyphenol	Thermo Fisher Scientific (Waltham, MA, USA	≈60%
Catechin (CA)	Polyphenol	Sigma Aldrich (St. Louis, MO, USA)	(HPLC) ≥ 96%
Resveratrol (Resv)	Polyphenol	Sigma Aldrich (St. Louis, MO, USA)	(HPLC) ≥ 98.5%
OMe-quercetin-glucuronide (3MQG)	Polyphenol metabolite	Donated	No information
Hippuric acid (HA)	Phenolic acid	Sigma Aldrich (St. Louis, MO, USA)	(titration by NaOH) 97.5-102.5%
3- (3′-hydroxyphenyl)propionic acid (3HPPA)	Phenolic acid	Alfa Aesar (Tewksbury, MA, USA)	(HPLC) ≥ 98%
3-hydroxypropionic acid (3HPA)	Phenolic acid	Sigma Aldrich (St. Louis, MO, USA)	(TLC) ≥ 99%
Quercetin-glucuronide (QG)	Polyphenol metabolite	Sigma Aldrich (St. Louis, MO, USA)	(LC/MS-ELSD) ≥ 95%
Malvidin-3-*O*-glucoside (MG)	Polyphenol	Millipore Sigma (Burlington, MA, USA) and Extrasynthese (Rhone, France)	(HPLC) ≥ 95%
3-hydroxybenzoic acid (3HBA)	Phenolic acid	Sigma Aldrich (St. Louis, MO, USA)	(HPLC) ≥ 98.5%

**Table 2 biomedicines-10-01264-t002:** CUS Schedule.

	Stress 1	Stress 2
Day	9:00 a.m.	4:00 p.m.
1	Crowd 1 h	Cold swim 5 m
2	Restrain 1 h	Lights on overnight
3	No food 7 h	Cage shake 20 m
4	Wet bedding 7 h	Restrain 1 h
5	Hot drier 10 m	Cage tilt 17 h
6	No water 7 h	Lights on overnight
7	Cold swim 5 m	Crowd 1 h
8	Restrain 1 h	No bedding 17 h
9	Cage shake 20 m	No food 17 h
10	Hot drier 10 m	Crowd 1 h
11	Cold swim 5 m	Restrain 1 h
12	Wet bedding 7 h	No bedding 17 h
13	Crowd 1 h	Cage tilt 17 h
14	Hot drier 10 m	Cage shake 1 h
15	Restrain 2 h	No food 17 h
16	Cage shake 1 h	Foot shock
17	Tail suspension 1 h	Cage tilt 17 h + Lights on
18	Cold swim 5 m	No bedding 17 h
19	Hot drier 10 m	Cage tilt 17 h + Lights on
20	Cage shake 1 h	No food + Lights on
21	Wet bedding 7 h	Restrain + Predator odor 1 h
22	Hot drier 10 m	Cage tilt 17 h
23	Crowd 2 h	Tail suspension 1 h
24	Restrain + Predator odor 2 h	Cage shake 1 h
25	Hot drier 10 m	Crowd 1 h
26	Cage shake 1 h	Lights on overnight
27	Restrain + Predator odor 1 h	No bedding 17 h
28	Hot drier 10 m	Cage tilt 17 h

## Data Availability

The authors declare that the data supporting the findings of this study are available within the paper and its Appendix A. Any remaining data that support the results of the study will be available from the corresponding author upon reasonable request.

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
