# Peer review of "Role of Polyphenol-Derived Phenolic Acid in Mitigation of Inflammasome-Mediated Anxiety and Depression"

_biomedicines, 2022, doi:10.3390/biomedicines10061264_

Round 1

Reviewer 1 Report

The manuscript entitled “Role of polyphenol-derived phenolic acid in mitigation of inflammasome-mediated anxiety and depression” was well conducted, structured and discussed. It contains a wide range of studies and it could be of interest for readers. The main concern is that the MG and the other phenolic compounds were purchased (synthetic), not isolated from a natural source as stated in some sections in the manuscript. This fact is really important as all results can be changed by the conformational structure of the phenolic compounds. Moreover, I have some comments for the authors.

Abstract

Line 27: Please, use the impersonal form.

Introduction

Line 46: Alzheimer disease is an example of neurodegenerative disease. Why did the other diseases not have an example?

Line 51: Please, add the meaning of IL and TNF the first time of appearing.

Material and methods

Line 125: Were the 13 natural compounds added separately? Which are those compounds? Which was the source of those compounds? Please, specify.

Sections 2.1.2 to 2.1.10: Why there is no reference of the methodology in these sections?

Line 141 and 142. How do you justify the difference in sample treatment (6h and 3h)?

Section 2.1.9: Briefly explain the methodology.

Line 220: Why was the MG purchased (synthetic origin)? In my opinion and in this type of research it is better if the nature of the antioxidants proceeds from natural sources, as we can find in the food. Additionally, the procedure of MG can significantly change all the obtained results.

Sections 2.2.3 and 2.2.4.  Please, numbered each sub-section for better understanding.

Results

Line 322, 441: Are you sure that the source of MG is natural? According to line 220 it was purchased.

Lines 330-331: In the phase “we pretreat murine primary cortical microglia cells with 10μM of the individual metabolites” I assume that they were added within an extract but when ingested these compounds are metabolized and their conformational structure changes. How can you overlap your results with the in vivo conditions? At least at this point it is unclear the goal of the pre-treatment.

Lines 352-356, 372-373: State that this was under your conditions and highlight that MG proceeds from a synthetic source.

Figure 5. Indicate the meaning of MG, CUS and EPM for better understanding.

Reviewer 2 Report

In Material and Methods, it is not clear the reason by which the authors have determined the cytotoxicity of MG at concentrations higher than 10 mM, because in the polyphenols screening in vitro, the concentration used was 10 μM. Moreover, at this concentration the activity was particularly high, according to the Figure 1.

Other question, and since in the other assays the concentrations used were …. and 25 or 100 μM, why the cytotoxicity of 10 μM was not performed? And why 100 μM was not tested?

In Figure 1, there are compounds that are not phenols: hippuric acid, 3-hydroxypropionic acid. Catecholamine, is an amine and not a phenol, although had a catechol strucuture. In addition, which catecholamine was used? Epinephrine, norepinephrine, dopamine???

MG was selected among 13 natural polyphenols…. Are they 13 or 10?

It is also known that only 5-10% of orally consumed polyphenols is absorbed in the small intestine, while 90-95% is further breakdown by gut microbiota fermentation in the colon or large intestine…… The authors reported that in the brain only 0.08 nM of MG was detected after 12.5 mg/Kg had been administered. This concentration would correspond to 25 μM (if considered 12.5 mg/L). In vitro, this concentration of MG presented activity, but 1 nM (0.001 μM) practically did not presented activity. How can the authors explain the activity found in vivo?

The question is made because in the Abstract, the authors have written:

 This study aims to elucidate the therapeutic potential of MG on inflammasome-25 induced inflammation in vitro and in a mouse model of chronic unpredictable stress (CUS). Here, 26 we show that MG is an anti-pyroptotic phenolic metabolite that targets, NLRP3, NLRC4, and AIM2 27 inflammasomes, subsequently reducing Caspase-1 and IL-1β protein levels in murine primary cor-28 tical microglia and, brain as also demonstrated its beneficial effect to counteract anxiety and depres-29 sion.

The present study supports the role of MG to mitigate bacterial-mediated inflammation (lipo-30 polysaccharide or LPS) in vitro and CUS-induced behavior impairment in vivo to address stress-31 induced inflammasome-mediated innate response

Round 2

Reviewer 1 Report

None.

Reviewer 2 Report

The work was improved, but it still needs some corrections.

Hippuric acid and 3-HPA are not phenols.

In our in vivo paradigm, the daily administration and accumulation of MG at 12.5mg/kg for 42 days seems to be enough to exert a biological function, as shown in our behavioral data. We might expect that the concentration of accumulated of MG after this long period may be higher than the concentration found in the brain, 0.08 nM, after 7 days of daily administration at 12.5mg/kg.

However, in our in vitro system, the concentration of 1 nM (closest to 0.08 nM found in the brain), was not potent enough to induce an inhibitory effect on inflammasome activation.

It is important to emphasize the difference between these two models or paradigms, and the difficulty to demonstrate the correlation between the two systems. Even so, both have been essential to examining the functional performance of MG, the toxicology and effectiveness on inflammasome activation in vitro, as also the potential anxiogenic and anti-depressive effect after submitting mice to chronic unpredictable stress.

This response is not enough and an explanation needs to be added in the manuscript. Without the information, the reader will do the same question I have made.